# Diagnostic and prognostic potential of eight whole blood microRNAs for equine sarcoid disease

Jeanne Cosandey[1], Eman Hamza[1]*, Vinzenz Gerber[1], Alessandra Ramseyer[1], Tosso Leeb[2], Vidhya Jagannathan[2], Klaudia Blaszczyk[1], Lucia Unger[1]

1 Swiss Institute of Equine Medicine, Department of Clinical Veterinary Medicine, Vetsuisse Faculty, University of Bern and Agroscope, Bern, Switzerland, 2 Institute of Genetics, Vetsuisse Faculty, University of Bern, Bern, Switzerland

* eman.hamza@vetsuisse.unibe.ch

**Data Availability Statement:** All relevant data are within the paper and its Supporting information files.

## Abstract

MicroRNAs have been proposed as biomarkers for equine sarcoids, the most prevalent equine skin tumors globally. This study served to validate the diagnostic and prognostic potential of whole blood microRNAs identified in a previous study for long-term equine sarcoid diagnosis and outcome prediction. Based on findings of a clinical examination at the age of 3 years and a follow-up following a further 5–12 years, 32 Franches-Montagnes and 45 Swiss Warmblood horses were assigned to four groups: horses with regression (n = 19), progression (n = 9), new occurrences of sarcoid lesions (n = 19) and tumor-free control horses (n = 30). The expression levels for eight microRNAs (eca-miR-127, eca-miR-432, eca-miR-24, eca-miR-125a-5p, eca-miR-134, eca-miR-379, eca-miR-381, eca-miR-382) were analyzed through reverse transcription quantitative polymerase chain reaction in whole blood samples collected on initial examination. Associations of sex, breed, diagnosis, and prognosis with microRNA expression levels were examined using multivariable analysis of variance. Sex and breed influenced the expression level of five and two microRNAs, respectively. Eca-miR-127 allowed discrimination between sarcoid-affected and tumor-free horses. No variation in microRNA expression was found when comparing horses with sarcoid regression and progression. Expression levels of eca-miR-125a-5p and eca-miR-432 varied in male horses that developed sarcoids throughout the study period in comparison to male control horses. While none of the investigated miRNAs was validated for predicting the prognosis of sarcoid regression / progression within young horses with this condition, two miRNAs demonstrated potential to predict if young male (though not female) tumor-free horse can develop sarcoids within the following years. Sex- and breed- biased miRNAs exist within the equine species and have an impact on biomarker discovery.

**Funding:** L.U. Grant number: 37900.1 IP-LS Innosuisse (https://www.innosuisse.ch) The funders had no role in study design, data collection and analysis, decision to publish, or preparation of the manuscript.

**Competing interests:** The authors have declared that no competing interests exists.

## Introduction

Equine sarcoids (ES) are the most globally prevalent skin tumors in equids [1, 2]. Bovine papillomavirus 1 and 2 (BPV 1 and 2) are considered as the major etiological agents of ES [3]. Genetic susceptibility appears to play an important role and is thought to have a polygenic base [4, 5].

Histopathology from a biopsy remains the gold-standard for diagnosis of ES, though is an invasive technique that can lead to aggravation of the condition [6]. Non-invasive methods comprise polymerase chain reaction (PCR) analyses for BPV 1 and 2 from swabs collected from the surface of putative ES lesions, together with clinical diagnosis (based on typical lesion appearance / localization and has approximately 80% sensitivity and specificity [7–10]). In challenging atypical cases, adjunct non-invasive tests can improve the reliability of ES diagnosis.

The clinical course of ES is unforeseeable: small, isolated occult or verrucous ES lesions can progress to substantial or multiple, nodular, fibroblastic, mixed or even malevolent (semi-malignant) lesions, or alternatively, such lesions spontaneously regress and disappear completely [11]. This makes the choice for or against therapy—resorting in a wait-and-see approach—extremely difficult. Furthermore, sarcoids can affect the value and use of an animal, and such unpredictable growth behavior can introduce a problematic uncertainty when selling such equids. Currently, no prognostic kit is available for prediction of the long-term outcome in ES disease.

MicroRNAs (miRNAs) are small non-coding 20–24 nucleotide-long RNA molecules that interact with target messenger RNAs (mRNAs), leading to mRNA degradation or translational repression, thus ultimately resulting in downregulation of target gene expression [12]. miRNA dysregulation can trigger disease manifestations and, consequently, miRNAs are reliable epigenetic biomarkers in humans for diverse disorders, including cancer [13, 14]. Circulating miRNAs have received particular attention as non-invasive liquid biopsies for early cancer detection, monitoring of cancer progression and assessment of prognosis [15–17]. Presently, the most commonly employed biofluids for circulating miRNA analyses are serum and plasma, while whole blood miRNAs have recently received increased research focus. Blood-cell specific miRNA expression has been identified within human peripheral blood and both, white and red blood cells contain substantial quantities of miRNAs [18–20]. In addition, whole blood contains a proportion of cell-free circulating miRNAs [21]. Interestingly, the mix of cellular and cell-free miRNAs allows for the detection of tumor-secreted miRNAs together with variations in miRNA expression secondary to the host immune responses to cancer [22].

Since 2009, over 60 studies have been published on miRNA expression within various equine physiological and pathological conditions, including ES disease [23]. Dysregulated miRNA expression profiles were observed in sarcoid tissue, BPV-transformed cell lines, serum and whole blood of ES-affected horses when compared to control samples [24–27]. Selected serum and whole blood miRNA candidates qualified as potential non-invasive diagnostic and prognostic biomarkers for ES disease [26, 28, 29].

An exploratory next-generation sequencing (NGS) study identified 14 potentially prognostic whole blood miRNAs allowing to discriminate between horses that showed regression vs. progression of ES lesions during further course of the disease [28]. This study aims to replace NGS(which is essentially an elaborate technique, and is unpractical for routine clinical diagnostics) with reverse transcription quantitative polymerase chain reaction (RT-qPCR), a less expensive and time-consuming miRNA quantification method for validation potential biomarkers in a larger, more diverse and independent study population [30]. Furthermore, this study investigated whether such proposed prognostic miRNAs are not only able to predict if

an ES affected horse can demonstrate disease progression, but also if a so far ES free horse may develop ES lesions within the next few years of life, and if they additionally have diagnostic potential.

## Materials and methods

### Study cohorts

Concerning this cohort study, clinical data were retrieved from longitudinal, large-scale health surveillance studies of the two most common Swiss equine breeds including 702 Franches-Montagnes (FM) and 1451 Swiss Warmblood (CH-WB) horses conducted a the Swiss Institute of Equine Medicine (ISME) [11, 31–33] (Fig 1). All horses had an initial clinical examination at the age of 3 years, when a whole blood sample was collected, together with a follow-up examination 5–12 years later.

Concerning all ES horses, the localization, number, and most clinically severe of ES lesion during the first and the follow-up examinations were recorded. Diagnosis of ES was based on clinical judgment of typical morphologic features and distribution pattern of ES lesions. From 2017 onwards, a validated diagnostic protocol was used to improve the accuracy of clinical diagnosis of ES disease. Only horses with a score of 15 or higher were classified as ES affected [34]. Cases were excluded from the study whenever relevant information was missing within clinical records, clinical diagnosis was equivocal, no initial blood sample was available, or whenever the horse was lost to follow-up. Additionally, all FM horses included in our exploratory NGS study were excluded from the current study to form an independent sample [28]. Age matched controls were randomly selected from the large pool of tumor-free horses. Attention was paid to reach a similar breed and sex distribution in the ES vs. CTL group (S2 Table).

Horses were categorized into four groups based on the clinical course of ES disease. The control (CTL) group comprised horses that were free of ES lesions at both the initial and follow-up examinations. Horses were assigned to the regression group (RGR) if they had ES lesions on initial examination, which completely disappeared by the time of follow-up. Horses were categorized as progression (PGR) group cases if they had ES lesions on initial examination, which deteriorated by the time of follow-up into a more severe clinical phenotype and/ or

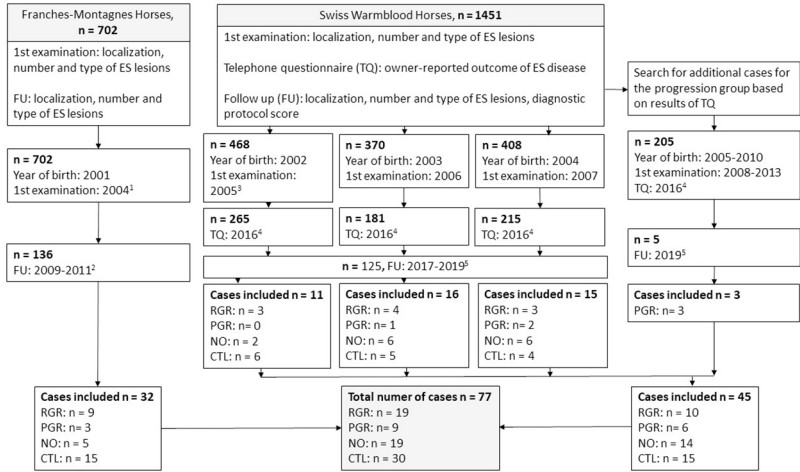

**Fig 1. Process of recruiting study subjects.** Abbreviations: TQ = telephone questionnaire, FU = follow-up examination, ES = equine sarcoid, RGR = regression, PGR = progression, NO = new occurrence, CTL = control, [1] = Mele et al., 2007, [2] = Berruex et al., 2016, [3] = Studer et al., 2007, [4] = Altermatt et al., 2021, [5] = Schöpfer, 2018.

a greater number of lesions. Horses were assigned to the new occurrence (NO) group if they were free of sarcoids on initial examination, though were affected by ES disease at the time of follow-up. Power-analysis calculations for each miRNA, identified as a potential prognostic biomarker for ES in our previous exploratory study, were based on a two-tailed t-distribution with a power of 0.9 and alpha of 0.05. The effect size was determined using the mean and standard deviation from the exploratory NGS study [28]. A minimum of 15–20 individuals were consequently defined as intended group size.

At the time of initial examination, blood samples were collected from the jugular vein into ethylenediamine tetraacetic acid tubes and stored at—80˚C within the ISME bioarchive. Horse owners were recruited for the above mentioned health surveillance study by the Franches-Montagnes and Swiss Warmblood breeding associations in collaboration with the ISME equine clinic [31, 32]. They gave verbal consent if they were willing to take part in the study and agreed to a blood sample being taken from their horses. Sample collection has been approved by the ethic committee of the Swiss Canton of Vaud (VD2227.0, VD2227.1 and VD2227.2).

## Pre-analytics

RNA was extracted according to an optimized protocol for long-term stored equine whole blood samples [35]. Prior to RNA-extraction, a synthetic spike-in control—cel-miR-39-3p [$5.6 \times 10^8$ copies, Cel_miR-39_1 miScript Primer Assay, Qiagen, Hombrechtikon, Switzerland] was added to each sample. This exogenous control allowed for the monitoring of RNA-extraction efficiency.

RNA concentration was measured using a Qubit 2.0 fluorimeter, in combination with the Qubit™ RNA BR assay kit [ThermoFisher, Reinach, Switzerland]. The RNA Integrity Number (RIN) was assessed using a Fragment Analyzer and the Standard Sensitivity RNA Analysis Kit [Advanced Analytical, Heidelberg, Germany]. The extracted RNA was stored at—80˚C for further analysis.

## Choice of candidate miRNAs

Ten out of 14 potentially prognostic miRNAs identified in the exploratory NGS study were selected for further RT-qPCR analysis [28]. Among such 10 miRNAs, seven miRNAs were selected (eca-miR-127, eca-miR-134, eca-miR-323-5p, eca-miR-379, eca-miR-381, eca-miR-382, eca-miR-432) that were recently identified as being encoded on equine chromosome 24 on a large miRNA cluster with putative tumor-suppressive function in ES disease [27, 28]. Additionally, three miRNAs that were not encoded by this cluster were also selected: eca-miR-107b, eca-miR-24 and eca-miR-125a-5p. Eca-miR-107b subjectively demonstrated the most prominent miRNA dysregulation profiles between horses with ES regression and progression, using the degPlot function from DEGreport from R [36]. Meanwhile, eca-miR-24 and eca-miR-125a-5p were—in contrast to all other selected miRNA candidates—upregulated in horses with ES progression compared to horses with regression [28].

Based on the results of the exploratory NGS study, the miRNA with the lowest coefficient of variation (CV) served as endogenous control, which was found to be eca-miR-30d (CV = 15.6) [28].

## RT-qPCR

Information on miRNAs primers and probes is depicted in S1 Table. The RT-reactions were performed using the TaqMan™ MicroRNA Reverse Transcription Kit [ThermoFisher, Reinach, Switzerland] and sequence-specific stem loop RT-primers for the candidate miRNAs as well as

endogenous and exogenous controls [ThermoFisher, Reinach, Switzerland]. Each RT-reaction contained 1 μL of extracted RNA, 0.5 μL of miRNA-specific stem-loop primer, 1.5 μL of 10x RT Buffer solution, 0.15 μL of 100 mM dNTP solution, 0.19 μL of RNAse inhibitor, 1 μL Multi-Scribe™ Reverse Transcriptase and 10.66 μL nuclease free water, for a total volume of 15 μL. The RT-reaction conditions were as follows: 30 min at 16˚C for primer annealing, 30 min at 42˚C for the extension phase and 5 min at 85˚C to stop the reaction with subsequent cooling at 4˚C.

Regarding the qPCR reactions, a total volume of 25 μL was used and consisted of 12.5 μL of 1x TaqMan™ Universal PCR Master Mix, no AmpErase™ UNG [ThermoFisher, Reinach, Switzerland], 0.625 μL of specific primer, 2.5 μL of RT product, and 9.375 μL nuclease free water. The qPCR reactions were run in duplicates in a 7300 Real-Time PCR system [Applied Biosystems, CA, USA] with a first denaturing step of 10 min at 95˚C, followed by 40 cycles consisting of denaturing at 95˚C for 15 sec, annealing and elongation at 60˚C for 60 sec, and a final inactivation step of 10 min at 99.9˚C. For assay quality assessment, the raw cycle quantification (Cq) values of the tested miRNAs were compared between the two technical replicates for each candidate miRNA and in each sample. A maximum Cq difference of 1 was accepted. If the difference was $> 1$, the reaction was repeated.

Efficiencies of all RT-qPCR assays were determined, using standard curves generated with 10-fold serial dilutions of synthetic analogues of the selected candidate miRNAs, the endogenous and exogenous controls [Microsynth AG, Balgach, Switzerland].

In this study, two differing data normalization methodologies were deployed.

Relative quantification of candidate miRNAs was performed using the $2^{-\Delta\Delta Cq}$ method [26, 37], with ΔCq being the Cq values of the candidate miRNA of interest or the endogenous control normalized to the exogenous control (cel-mir-39-3p) and ΔΔCq being the ΔCq value of the candidate miRNA normalized to the ΔCq of the endogenous control (eca-miR-30d).

In addition, normalized copy numbers of the candidate miRNAs were calculated using standard curves. Briefly, standard curves were generated for the candidate miRNAs using six 10-fold serial dilutions of synthetic miRNA [Microsynth AG, Balgach, Switzerland]. The Cq values for each dilution were plotted against the corresponding copy numbers in a 10-log based graph. A linear regression equation was created to extrapolate the slopes ($m$) and intercepts ($b$) from the standard curves. Such values were used to calculate the absolute copy number of the individual miRNAs candidates and controls, using the following formula: $10^{\frac{Cq-b}{m}}$ [37]. The obtained copy numbers of the candidate miRNAs and the endogenous control were normalized to the copy number of the exogenous control. Finally, the normalized copy number of the candidate miRNA was normalized to the normalized copy number of the endogenous control, yielding the absolute copy number of the specific candidate miRNA for each corresponding sample. This normalization method was chosen in addition to the $2^{-\Delta\Delta Cq}$ method in order to take into account the efficiencies of RT-qPCR reactions for each investigated miRNA.

Prior to running samples from the study cohort, RNA eluates from the previous NGS study were evaluated using RT-qPCR [28]. The $2^{-\Delta Cq}$ values of each candidate miRNA were consequently compared to the corresponding normalized read counts reported in the NGS study, using Pearson correlation coefficients. This enabled determination of the concordance for both miRNA quantification methods (for each of the candidate miRNAs). Candidate miRNAs with a positive correlation between both methods were preferentially selected for further RT-qPCR analyses in this study.

## Statistical analysis

Statistical analysis was performed using NCSS 12 [NCSS, Kaysville, Utah, USA].

The distribution of breed and sex in all four groups were compared using Fisher's exact test. Stallions (m = male) and geldings (mc = male castrated) were assigned as male horses and compared to mares (f = female).

miRNA expression data were not normally distributed and were therefore log-transformed. Outliers for each candidate miRNA were detected using Grubb's test. Whenever outlier data points for at least one candidate miRNA were identified in a sample, the sample was excluded from further analysis.

Multivariable analysis of variance (MANOVA) was performed to determine the effect of breed, sex, diagnosis, prognosis and their interactions on the expression of candidate miRNAs. Three MANOVA models were set up. The first model assessed the diagnostic potential of the candidate miRNAs and whether this was influenced by breed and sex. In this model, miRNA expression was compared between the groups of horses with ES lesions (RGR + PGR) and without ES lesions (NO + CTL) at the time of initial examination. The second model evaluated the potential of miRNA to predict the outcome in horses with ES-lesions (RGR vs. PGR). The third model examined the prognostic potential of candidate miRNAs to predict novel appearances of ES within presently ES-free horses (NO vs. CTL) and whether this was influenced by breed and sex. MANOVA involves two analytical steps, multi-variate and univariate. The first examines the effect of each single variable and their interactions on the common expression of all analyzed candidate miRNAs; with p-values ≤ 0.05 considered significant. The second step performed an identical analysis to multi-variate, though on an individual candidate miRNA. For each of the eight candidate miRNAs, p-values ≤ 0.05 were adjusted for multiple testing by the Benjamini-Hochberg method and considered significant if they were smaller than the adjusted p-value. When significance was achieved in MANOVA, the expression of the corresponding miRNA was further compared between the groups using Mann-Whitney U test or, when more than two groups were compared, using One Way analysis of variance with Kruskal-Wallis Multiple-comparison Z-value Test (Dunn's Test) with Bonferroni correction. Z-values > 2.6383 indicated a significant difference.

Regarding miRNAs with significant variations in expression between groups, a receiver operating characteristic (ROC) curve was plotted to identify the area under the curve (AUC) and to calculate their specificity and sensitivity as diagnostic and/ or prognostic whole blood biomarkers. Additionally, the positive likelihood ratio (LR+), negative likelihood ratio (LR-), diagnostic odd ratio (DOR), pre-test probability (PRP) and post-test probability (POP) were reported [38]. P-values ≤ 0.05 were considered significant.

## Results

### Study cohort selection

The recruiting process for the study populations and information on the experimental timeline is illustrated in Fig 1. All FM horses were born in 2001, initially examined in 2004, and had a clinical follow-up between 2009 and 2011. CH-WB horses were born between 2002 and 2010, had their first examination between 2005 and 2013, and the clinical follow-up was performed between 2017 and 2019. The time between examinations was a mean ± SD (range) of 5.8 years ± 0.42 (5–7) for FM horses and 10.9 years ±1.3 (6–12) for CH-WB horses.

A flow diagram depicting the recruitment of FM horses has been previously published [11] and further information is given in S1 Fig. For CH-WB horses, the process of case recruitment and reasons for nonparticipation at each stage are given in S2 Fig. In total, 32 FM horses and

45 CH-WB horses were recruited, of which 19 met the inclusion criteria for all RGR-group, 9 for the PGR group, 19 for the NO group and 30 for the CTL group (Table 1). The minimum-required group size was achieved for the RGR, NO and CTL groups, though only nine horses met the inclusion criteria for the PGR group. One CH-WB horse within the PGR group was euthanized due to progression of ES lesions prior to the follow-up examination. In this case, a detailed description of ES lesions prior to euthanasia was provided by the attending veterinarian.

An overview of the biological variables and clinical data of all horses included in the study is presented in S2 Table. The sex and breed distributions did not significantly differ among the RGR, PGR, NO and CTL groups (p = 0.27 and p = 0.35, respectively) and are illustrated in Table 1.

## Pre-analytics

RNA quantity was too low to be determined in five samples. Regarding the remaining 72 samples, the mean ± SD (range) RNA quantity was 29.7 ng/μL ± 15.98 (6–76.8) and RIN was 2.55 ± 1.76 (1–10) (S3 Table).

## RT-qPCR

When evaluating the correlation between NGS and RT-qPCR results for each candidate miRNA, Pearson correlation coefficients were positive for seven miRNAs, though this was negative for eca-miR-107b, eca-miR-381 and eca-miR-323-5p (S3 Fig). miRNAs with negative correlations were excluded from further analysis, except for eca-miR-381, as it was the top-ranking differentially expressed miRNA in the exploratory NGS-study and was consequently considered worthwhile for further testing [28]. This reduced the number of candidate miRNAs for RT-qPCR analysis to eight.

The raw Cq values and the $2^{-\Delta\Delta Cq}$ values, as well as normalized copy numbers of the eight analyzed candidate miRNAs are depicted in S3 and S4 Tables.

The two methods used for data normalization demonstrated a good correlation (S5 Table). In addition, for all miRNAs, the RT-qPCR efficiencies were greater than 99%. Hence, only the results of the $2^{-\Delta\Delta Cq}$ method are described here. When present, variations in miRNA expression between the two methods were specifically indicated.

Four samples were excluded from further analysis as they were showing one (or more) outlier data points using Grubb's test (S3 Table).

## Influence of biological variables on miRNA expression

All results obtained with MANOVA analysis are depicted in S6 Table. The multivariate analysis demonstrated a significant effect of sex (p = 0.019) and breed (p = 0.002) on miRNA expression. The univariate analysis for sex revealed a significant influence on the expression of eca-miR-127 (p < 0.001, p adj = 0.006), eca-miR-134 (p = 0.002, p adj = 0.031), eca-miR-379 (p < 0.0001, p adj = 0.019), eca-miR-382 (p < 0.0001, p adj = 0.025) and eca-miR-432 (p = 0.0001, p adj = 0.013). Regarding these five candidates, miRNA expression was up-regulated in male, rather female, horses. The influence of breed was significant for eca-miR-134 (p = 0.014, p adj = 0.019) and eca-miR-382 (p = 0.002, p adj = 0.006). In addition, the influence of breed was significant for eca-miR-381, although, only when using the normalized copy number method (p = 0.008, p adj = 0.019). Regarding all three breed-biased miRNAs, the expression was up-regulated in FM horses in comparison to CH-WB horses.

Table 1. Sex and breed distribution among all four clinical groups of the study cohort.

| | RGR | PGR | NO | CTL |
|---|---|---|---|---|
| **Repartition of sex within the groups** (p = 0.27) | | | | |
| **Mares** | 12/19 (63.2%) | 4/9 (44.4%) | 7/19 (37,8%) | 17/30 (56.6%) |
| **Geldings** | 7/19 (37,8%) | 5/9 (55,6%) | 12/19 (63.2%) | 11/30 (36.7%) |
| **Stallions** | 0/19 (0.0%) | 0/9 (0.0%) | 0/19 (0.0%) | 2/30 (6.7%) |
| **Repartition of breed within the groups** (p = 0.35) | | | | |
| **Franches-Montagnes** | 9/19 (47.4%) | 3/9 (33,3%) | 5/19 (26.3%) | 15/30 (50%) |
| **Swiss warmblood** | 10/19 (52.6%) | 6/9 (66,7%) | 14/19 (73,7%) | 15/30 (50%) |

p-values refer to variations of sex and breed distribution among groups. Abbreviations: RGR = regression, PGR = progression, NO = new occurrence, CTL = control.

## Diagnostic miRNAs

Within the multivariate analysis, a significant effect of diagnosis on miRNA expression was found (p < 0.001). However only eca-miR-127 demonstrated significantly different expression in the ES affected (RGR + PGR) vs. ES free groups (NO + CTL) at the time of initial examination (p = 0.002, p adj = 0.006) (Fig 2A) using univariate analysis. The ROC curve analysis indicated an AUC of 0.64 (CI = 0.50–0.75, p = 0.01), with a sensitivity of 59% (CI = 39–78%) and a specificity of 64% (CI = 49–77%) for whole blood eca-miR-127 in diagnosing ES within the entire study population (Fig 3A). The LR+ and LR− for eca-miR-127 in diagnosing ES were 1.64 and 0.64, respectively, with a DOR of 2.5, a PRP of 0.37, and a POP of 0.48. The interaction between sex and diagnosis had no significant effect on the expression of eca-miR-127, indicating that sex has no influence on the use of this miRNA as a diagnostic biomarker. Conversely, breed significantly influenced the use of eca-miR-127 as a diagnostic biomarker for ES (p < 0.0001, p adj = 0.006). While eca-miR-127 expression was significantly up-regulated in ES affected, in comparison to ES free FM horses (Z = 3.37), such a difference could not be confirmed for CH-WB horses (Fig 2B). The ROC curve analysis for whole blood eca-miR-127 as a diagnostic biomarker for ES disease in FM horses indicated an AUC of 0.84 (CI = 0.61–0.94, p < 0.0001), with a sensitivity of 91% (CI = 59–99%) and a specificity of 83% (CI = 59–96%)

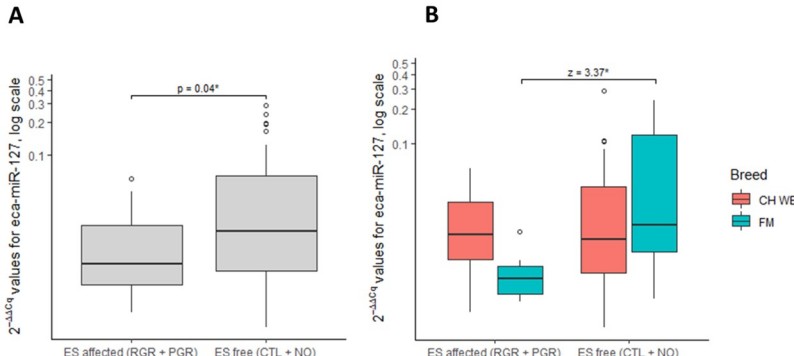

**Fig 2. Box plots depicting relative whole blood expression of eca-miR-127 in ES affected vs. ES free horses in general (A) and post-separation by breed (B).** The y-axis demonstrates the $2^{-\Delta\Delta Cq}$ values of eca-miR-127 on a log scale. Significant p and Z-values are marked with an asterisk (*). In Fig 2A, the p-value of Mann-Whitney U test was reported. In Fig 2b, the Kruskal-Wallis Multiple-Comparison Z-Value (Dunn's Test) with Bonferroni correction was reported. Abbreviations: RGR = regression, PGR = progression, NO = new occurrence, CTL = control, ES = equine sarcoid.

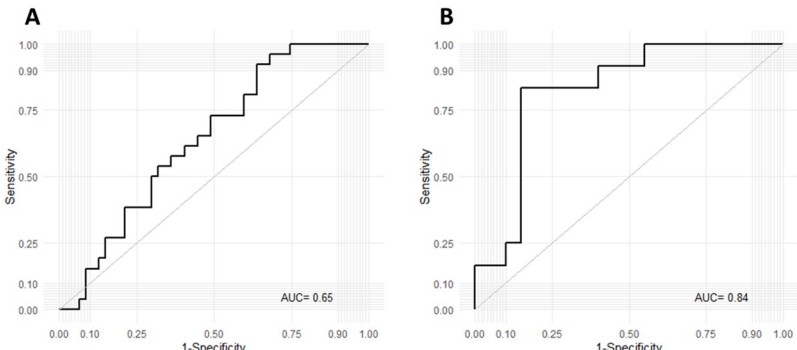

**Fig 3. ROC curve analysis assessing whole blood expression of eca-miR-127 for discrimination of ES affected (RGR + PGR) from ES-free horses (NO + CTL) at the time of initial examination for the entire population (A) and solely within FM horses (B).** Abbreviations: ROC = Receiver Operating Characteristic. AUC = Area Under the Curve, RGR = regression, PGR = progression, NO = new occurrence, CTL = control, FM = Franches-Montagnes horses.

(Fig 5B). The LR+ and LR− for eca-miR-127 in diagnosing ES in FM horses were 5.45 and 0.11, respectively, with a DOR of 50, a PRP of 0.38 and a POP of 0.77.

### Prognostic miRNAs

No variation was found when comparing miRNA expression between the RGR and PGR groups (p = 0.78) in the multivariate analysis. The influence of sex and breed as well as their interactions with prognosis in these groups could not be evaluated due to the small number of individuals within the PGR group.

Comparing miRNA expression between the CTL and NO groups, no significant difference was found (p = 0.06) in the multivariate analysis. However, this study found that the interaction of prognosis and sex was significant (p = 0.005). This was confirmed for eca-miR-127 (p = 0.002, p adj = 0.01), eca-miR-379 (p = 0.001, p adj = 0.006), eca-miR-432 (p = 0.002, p adj = 0.02) and eca-miR-125a-5p in the univariate analysis. Regarding the latter, a significant difference was only identified through the $2^{-\Delta\Delta Cq}$ method (p = 0.02, p adj = 0.03). The expression of the four sex-biased miRNAs was further compared between the CTL and NO groups following sex separation, using Kruskal-Wallis Multiple-Comparison Z-value with Bonferroni correction (Fig 4). A variation in miRNA expression between the NO vs. CTL groups was only confirmed for male horses for eca-miR-125a-5p (Z = 3.05) (Fig 4A), and eca-miR-432 (Z = 2.66) (Fig 4D), though not for eca-miR-127 (Z = 2.44) (Fig 4B) and eca-miR-379 (Z = 1.64) (Fig 4C). Regarding female horses, no significant variations in miRNA expression between CTL and NO groups could be found. Furthermore, there was a significant variation in miRNA expression between male and female horses in the CTL group, for eca-miR-127 (Z = 3.81) (Fig 4B), eca-miR-379 (Z = 3.92) (Fig 4C) and eca-miR-432 (Z = 3.66) (Fig 4D).

Regarding the prediction of new occurrence of ES lesions, ROC curve analyses were performed only for male horses for eca-miR-125a-5p, and eca-miR-432. Concerning such two miRNAs, the complete results of the ROC curve analyses are depicted in Table 2 and Fig 5.

### Discussion

This study evaluated previously identified diagnostic and prognostic whole blood biomarker candidates for ES, that can be useful for future clinical applications. However, breed- and sex-

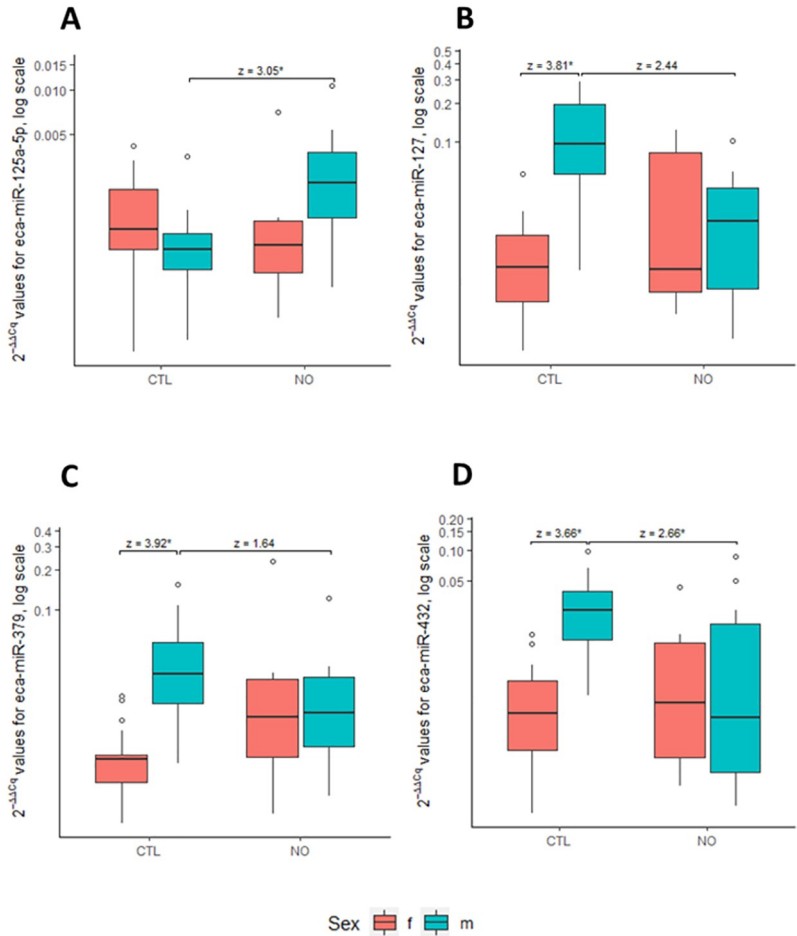

**Fig 4. Box plots depicting relative whole blood expression of eca-miR-125a-5p (A), eca-miR-127 (B), eca-miR-379 (C) and eca-miR-432 (D) between the control and new occurrence groups, separated by sex.** The y-axis demonstrated the $2^{-\Delta\Delta Cq}$ values of the microRNA candidate on a log scale. The Kruskal-Wallis Multiple-Comparison Z-value (Dunn's Test) with Bonferroni correction was reported and marked with an asterisk (*) when significant. Abbreviations: CTL = control, NO = new occurrence, m = male, f = female.

specific variations in equine miRNA expression complicate the validation of universally valid biomarker candidates.

The samples utilized in this study were derived from a population-based biobank built up over years. A limitation to this approach was that the sample pool was restricted, whereby in

**Table 2. ROC curve analysis for miRNAs as prognostic biomarker in the prediction of sarcoid onset in healthy 3-year old male horses.**

| | AUC (95% CI) | P value | Sensitivity (%) (95% CI) | Specificity (%) (95% CI) | LR+ LR- | DOR | PRP | POP |
|---|---|---|---|---|---|---|---|---|
| **eca-miR-432** | 0.76 (0.47–0.91) | 0.006 | 67 (35–90) | 75 (43–95) | 2 0.50 | 4 | 0.5 | 0.73 |
| **eca-mir-125a-5p** | 0.84 (0.56–0.95) | 0.0001 | 83 (52–98) | 83 (52–98) | 5 0.64 | 7.86 | 0.5 | 0.83 |

Abbreviations: AUC = Area under the curve, CI = confidence interval, LR+ = positive likelihood ratio, LR- = negative likelihood ratio, DOR = diagnostic odd ration, PRP = pre-test probability, POP = post-test probability.

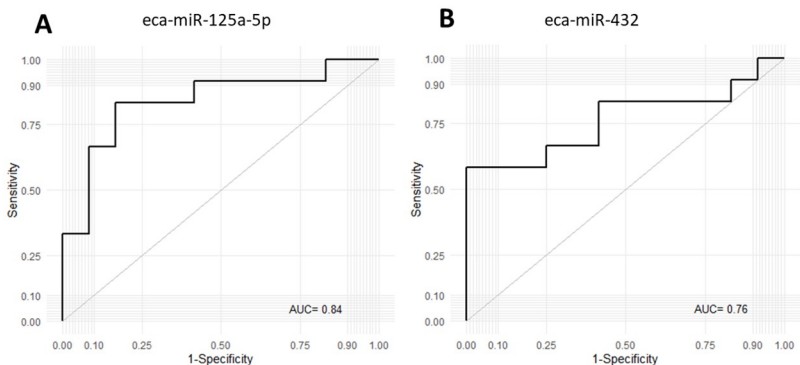

**Fig 5. ROC curve analyses assessing whole blood expression of eca-miR-125a-5p (A) and eca-miR-432 (B) for discrimination of male horses, that developed ES lesions throughout the study period (male NO) and horses, that remained tumor-free (male CTL).** Abbreviations: ROC = Receiver Operating Characteristic. AUC = Area Under the Curve, NO = new occurrence, CTL = control, ES = equine sarcoid.

one group the required sample size calculated for sufficient statistical power could not be met. However, employing such biobanked samples allowed for the realization of this elaborate longitudinal study while avoiding prolonged prospective sampling and particularly arduous follow-up periods spanning over years, though which are essential to assess the clinical course of ES. Sporadic appearance of ES lesions is reported up to an age of 31 years [26, 39]. However, ES lesions rarely develop in horses older than 7 years [40–43]. Even if the observation period in this study did not cover the entire lifespan of the study subjects, all horses were at least monitored up to an age of 8–15 years, which—in itself—constitutes a unique and valuable study cohort for ES disease. One limitation in this study is the lack of histopathological confirmation of ES disease, which is considered a diagnostic gold standard [44]. At the time of initial examination, biopsies were not performed since they represent a risk for disease aggravation, as any kind of trauma. Furthermore, in none of the horses, a surgical excision of the relatively mild ES lesions was indicated. In order to offset the lack of histopathological confirmation, only cases with a typical clinical presentation were included, and since its availability in 2017, an improved diagnostic protocol was additionally applied. Since it has been previously demonstrated that atypical cases predominantly lead to mis-classification, the reliability of diagnosis in our study cohort was deemed high [9].

Whole blood has several advantages compared to traditional biofluids used for circulating miRNA analysis (most importantly serum and plasma). The miRNA quantity in whole blood is markedly higher, which consequently allows for the detection of low-level expressed miRNAs [22, 29]. Furthermore, hemolysis in serum samples secondary to collection and processing was found to have a significant effect on equine serum miRNA expression profiles and could complicate miRNA analysis [26, 29]. Conversely, the use of whole blood bypasses potentially error-prone steps of preanalytical sample handling, since whole blood can be used directly as primary sample without further processing [45]. The aim of this study was to correlate whole blood miRNA expression at the beginning of the study, with the long-term outcome of ES disease. This implied the implementation of long-term stored bioarchive samples without RNA stabilizing additives. However, optimized sample handling and RNA extraction enabled the recovery of sufficient quantities of RNA with adequate quality for robust miRNA investigation, as demonstrated in this group's previous studies [28, 35]. RT-qPCR was employed for miRNA analysis in this study, since it shows excellent sensitivity, specificity, and reproducibility, requires only widely available standard equipment, and is thus more suitable

for routine use than NGS [30]. Relative miRNA quantification methods are most commonly applied when using RT-qPCR [46]. The two normalization methods used in this study demonstrated overall good agreement. Regarding data normalization, the choice of appropriate reference genes is of particular importance in order to avoid bias in miRNA quantification. There is, however, neither a universally valid endogenous control nor a consensus on the most appropriate references miRNAs in biofluids [46, 47]. We based our choice on miRNA expression data from the exploratory NGS study, which perfectly reflected the nature of the samples included in the current validation study [28, 46]. Eca-miR-30d was the only miRNA in our dataset that qualified as endogenous control. Thus, we used this miRNA as a single universal normalizer, even if a combination of several normalizers is preferable and more reliable [48]. In future, novel, absolute miRNA quantification methods that do not require the use of an endogenous control, such as the miRNA enzyme immunoassay (miREIA) technique, will be considered [30].

One out of eight tested miRNAs demonstrated potential as a diagnostic biomarker for ES disease. Eca-miR-127 was downregulated in ES affected horses when compared to CTL horses though had a poor sensitivity and specificity as a universal diagnostic biomarker due to a breed effect. Similar results in terms of diagnostic utility have been recently reported for serum eca-miR-331. Whereas this particular serum miRNA was not influenced by biological variation among study subjects, breed had a significant influence on the use of whole blood eca-miR-127 as a diagnostic biomarker in the current study [26]. Eca-miR-127 was not a diagnostic discriminator between ES affected and ES free CH-WB horses, though in FM horses, it had a better diagnostic sensitivity and specificity for ES disease than clinical diagnosis [9]. In concordance with previous studies, this highlights that breed can be a factor of considerable influence on the expression of equine miRNAs and has to be carefully taken into account as a potential confounding factor for miRNA biomarker discovery [49, 50].

In human medicine, multiple studies have investigated the potential of miRNAs to predict the outcome of cancer and an increasing number of meta-analyses recently confirmed selected prognostic miRNA candidates mainly in tumor tissue, and to a lesser extent in blood circulation [51–57]. However, some studies show that biological factors such as ethnicity can influence the potential of miRNA for long-term prediction of cancer outcome and that the prognostic potential of selected miRNAs can only be confirmed at the tumor tissue level, though not in blood circulation [51, 56, 58]. In our study, none of the eight tested whole blood miRNAs was able to predict the outcome of ES disease in horses that already had ES lesions at the time of blood sample collection. Thus, the results of the exploratory NGS study could not be confirmed [28]. Discrepancies of cancer-specific miRNA signatures among studies are often reported and may be partly due to the fact that methods for miRNA detection and quantification are not standardized [59]. In our study, the correlation of NGS and RT-qPCR results was moderate to excellent for the majority of the analyzed miRNAs, although we found negative correlations for three miRNAs [28]. Still, it seems unlikely that the different methods used in the exploratory and validation studies are the only reason for such discrepancies. Likely, one main cause for the differing results is the small sample size used in the exploratory NGS study [28]. This is a common approach in the biomarker discovery phase, though results in limited statistical power and bears the risk that several miRNA candidates may not be confirmed in larger-scale validation studies [60]. Additionally, for this study, only a small group of horses with progression of ES lesions could be recruited. This reduced the chance of detecting a true effect and hampered investigations for the interaction of breed and sex with the use of the proposed miRNAs as prognostic biomarkers.

Furthermore, this study probed if the potentially prognostic miRNAs identified in the exploratory NGS study were able to predict if young ES free horses would develop ES lesions

later in life. Eca-miR-432 and eca-miR-125a-5p were able to prognosticate NO of ES lesions, however, only in male and not in female horses. In human medicine, the homologues of these two equine miRNAs, namely hsa-miR-432 and hsa-miR-125a-5p, were shown to be prognostic serum biomarkers for various neoplastic conditions, including breast, and lung cancer [61–63]. In horses, eca-miR-432 is part of a large, reportedly tumor-suppressive miRNA cluster on the equine chromosome 24 [27, 28]. It was found to be upregulated in ES lesions and downregulated in BPV-transformed equine fibroblasts, in whole blood of horses with ES RGR vs. PGR, and in the current study in whole blood of ES affected vs. ES-free horses and in male horses with NO of ES lesions vs. male CTL horses [24, 25, 27, 28]. These findings suggest that eca-miR-432 might be selectively secreted into the circulation [27]. Our research group (and other groups) have previously found that only a small portion of miRNAs is released from the tumor into the extracellular space and that in some instances opposing trends of miRNA expression levels can be observed in these two compartments [29, 64, 65]. This might be partly explained by the ability of tumor cells to actively absorb miRNAs from the circulation [66]. Additionally, circulating miRNAs are shed from stromal cells, and exert either oncogenic or tumor-suppressive effects in the tumor microenvironment [67, 68]. The confirmation of dysregulation for eca-miR-432 in several studies suggests that this miRNA might represent central regulatory elements in ES disease, although its specific biological functions need to be investigated further.

We observed that a marked effect of sex must be taken into account in the use of eca-miR-432 and eca-miR-125a-5p as prognostic biomarkers for ES disease. miRNA expression can be significantly influenced by sex steroid hormones and chromosome X-linked genes within invertebrates and vertebrates [69, 70]. Sex-specific differences in miRNA expression have also been observed in various pathological conditions in humans, such as in autoimmune, endocrine, or neoplastic disorders [71–73]. This is the first study to highlight sex-biased miRNA expression in horses, with male horses demonstrating up-regulation of several miRNAs in whole blood, when compared to female horses. Even if ES disease is, as such, not a sexually dimorphic disease, ignoring sex-biased miRNA expression can compromise results and consequently the identification of male or female specific miRNA biomarker candidates [73]. Further validation of sex-specific miRNA biomarker panels for clinical applications in equine medicine is warranted.

## Conclusion

This study found further evidence that whole blood miRNAs can represent potential diagnostic and prognostic biomarkers for ES disease. However, due to the significant influence of biological variation on miRNA expression, no universally valid miRNA biomarker currently qualifies for clinical applications. Additionally, miRNAs are not yet seeing clinical use as biomarker in human medicine, due to lack of assay standardization and reproducibility of results among studies [46]. Nevertheless, it is warranted to further explore the proposed diagnostic and prognostic miRNA biomarkers for ES disease in larger study cohorts. These epigenetic biomarkers can take us one step further towards personalized veterinary medicine by successfully monitoring disease progression or predicting response to therapy in the near future [74].

## Supporting information

**S1 Table. Sequences of the candidate miRNAs and their stem-loop primers and information on TaqMan™ MicroRNA Assay kits used for RT-qPCR.**
(DOCX)

**S2 Table. Overview study cohort.** All horses were three years old at the time of initial examination. Information on biological variables (breed, sex) and time between initial and follow-up examination is provided. For ES affected horses, information on worst ES type as well as localization and number of ES lesions at both examinations and on the type of therapy applied is given. Abbreviations: ES = Equine sacoid, ID = identity, FM = Franches-Montagnes, WB = Swiss Warmblood, NA = not applicable, f = female, m = male, mc = male castrated. (DOCX)

**S3 Table. Results of relative miRNA quantification.** Outlier samples detected with Grubb's test are marked with the sign "*". Abbreviations: RQ = relative quantification using $2^{-\Delta\Delta Cq}$ method, NormCpN = normalized copy number (absolute quantification), Cq = quantification cycle, CTL = control, RGR = Regression, PGR = Progression, NO = New occurrence, RIN = RNA integrity number. (XLSX)

**S4 Table. Raw Cq-values of all candidate miRNAs and endogenous and exogenous controls for each sample.** For technical reasons, the measurements of miRNA candidate with RT-qPCR were conducted in 3 subsequent parts. For each part, the endogenous and exogenous control were included to assure optimal normalization of the results. Raw Cq-values of each part are reported in a separated column. Abbreviations: Cq = quantification cycle. (XLSX)

**S5 Table. Pearson correlation coefficients between the $2^{-\Delta\Delta Cq}$ and the normalized copy number methods of normalization for the eight final miRNA candidates.** Abbreviations: Cq = quantification cycle, CI = Confidence interval. (DOCX)

**S6 Table. Results of the MANOVA analysis for the 3 models.** The first model (A) assessed the diagnostic potential of the candidate miRNAs, the second model (B) evaluated the potential of miRNA to predict the outcome in horses with ES-lesions and the third model (C) examined the prognostic potential of the candidate miRNAs to predict new appearance of ES. (DOCX)

**S1 Fig. Selection of study subjects and reasons for nonparticipation in the cohort of Franches Montagnes horses.** (PDF)

**S2 Fig. Selection of study subjects and reasons for nonparticipation in the cohort of Swiss Warmblood horses.** (PDF)

**S3 Fig. Pearson correlation coefficients between previously published normalized reads counts from the exploratory NGS study and their corresponding $2^{-\Delta Cq}$ obtained after RT-qPCR from the same RNA eluates for the selected miRNA candidates.** (PDF)

## Acknowledgments

We thank Dr. Dominik Burger and all colleagues and horse owners who have contributed to the collection of phenotype information and samples that were used in this study.

## Author Contributions

**Conceptualization:** Jeanne Cosandey, Klaudia Blaszczyk, Lucia Unger.

**Data curation:** Jeanne Cosandey.

**Formal analysis:** Jeanne Cosandey, Eman Hamza.

**Funding acquisition:** Lucia Unger.

**Investigation:** Jeanne Cosandey, Lucia Unger.

**Methodology:** Jeanne Cosandey, Eman Hamza, Tosso Leeb, Vidhya Jagannathan, Lucia Unger.

**Project administration:** Lucia Unger.

**Resources:** Alessandra Ramseyer, Klaudia Blaszczyk.

**Supervision:** Eman Hamza, Vinzenz Gerber, Lucia Unger.

**Validation:** Eman Hamza.

**Writing – original draft:** Jeanne Cosandey, Lucia Unger.

**Writing – review & editing:** Eman Hamza, Vinzenz Gerber, Alessandra Ramseyer, Tosso Leeb, Vidhya Jagannathan, Klaudia Blaszczyk, Lucia Unger.

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
