## [Decision Letter · Decision Letter 0]

4 Aug 2021

PONE-D-21-19643

Diagnostic and prognostic potential of eight whole blood microRNAs for equine sarcoid disease

PLOS ONE

Dear Dr. Cosandey,

Thank you for submitting your manuscript to PLOS ONE. After careful consideration, we feel that it has merit but does not fully meet PLOS ONE’s publication criteria as it currently stands. Therefore, we invite you to submit a revised version of the manuscript that addresses the points raised during the review process.

We look forward to receiving your revised manuscript.

Kind regards,

Silvia Sabattini

Academic Editor

PLOS ONE

2. PLOS requires an ORCID iD for the corresponding author in Editorial Manager on papers submitted after December 6th, 2016. Please ensure that you have an ORCID iD and that it is validated in Editorial Manager. To do this, go to ‘Update my Information’ (in the upper left-hand corner of the main menu), and click on the Fetch/Validate link next to the ORCID field. This will take you to the ORCID site and allow you to create a new iD or authenticate a pre-existing iD in Editorial Manager. Please see the following video for instructions on linking an ORCID iD to your Editorial Manager account: https://www.youtube.com/watch?v=_xcclfuvtxQ.

Reviewers' comments:

Reviewer's Responses to Questions

**Comments to the Author**

1. Is the manuscript technically sound, and do the data support the conclusions?

Reviewer #1: Partly

Reviewer #2: Yes

2. Has the statistical analysis been performed appropriately and rigorously? 

Reviewer #1: Yes

Reviewer #2: Yes

3. Have the authors made all data underlying the findings in their manuscript fully available?

Reviewer #1: Yes

Reviewer #2: Yes

4. Is the manuscript presented in an intelligible fashion and written in standard English?

Reviewer #1: No

Reviewer #2: Yes

5. Review Comments to the Author

Reviewer #1: The article by Unger et al. investigates two different aspects of miRNA in sarcoids. The authors already have several publications in this field and the aims and results of this study are partially based on previously published material.

In my opinion the results provide important guidance for following researchers therefore I am in favour for publication, however see below major concerns regarding the inclusion criteria.

Firstly, the authors evaluate the replacement of NGS with less expensive RT-qPCR for miRNA quantification in whole blood samples, demonstrating that RT-qPCR is suitable for this analysis showing higher sensitivity, specificity and reproducibility. This is an important achievement that may significantly facilitate future research in this field.

Secondly, the authors evaluate the prognostic role of miRNA in predicting sarcoid recurrence/de novo occurrence including a reasonable number of horses with a long-term follow up. With respect to this section of the study, the inclusion criteria may have significantly affected the obtained results.

While the diagnosis of sarcoid subtypes is clinical, the gold standard for the diagnosis of ES is histopathology. Cases included in the study were diagnosed by clinical examination while cases with an uncertain clinical diagnosis were excluded. As the sensitivity for the clinical diagnosis, as correctly reported by the authors, is about 80%, there is a potential up-to-20% of horses that do not have sarcoids: this may significantly alter the results of the study.

Also, I understand that the improved protocol developed in 2017 was not applied to diagnose ES in this study as the last round of first examination was performed in 2013 according to Fig. 1. Is this correct?

Since these miRNA biomarkers may then be used for future research it is crucial to include cases with a gold-standard-diagnosis.

Additional minor comments:

Line 94- sentence incomplete?

Line 367- rephrase

Line 244- provide reference

Reviewer #2: The study is well thought, well written and followed a scientifically correct method. The statistical analysis is appropriate. The topic of the study is of absolute interest in veterinary oncology an perfectly in line with novel biomolecular tools in this discipline. I really congratulate with authors and thank them for this important improvement of knowledge.

Some minor points:

line 41 and others following: please put a space bar between text and brackets

line 94: maybe a typo

line 108: were control horses of the same breed of the studied ones? and of the same age? please specify the inclusion criteria con control cases.

Why didn't the authors took in consideration the histotype of the sarcoids? Even if not so relevant from a clinical point of view, it would be interesting to evaluate is any difference could be present among different histotypes and relative miRNAs.

6. PLOS authors have the option to publish the peer review history of their article (what does this mean?). If published, this will include your full peer review and any attached files.

Reviewer #1: No

Reviewer #2: No

---

## [Author Response · Author response to Decision Letter 0]

28 Oct 2021

Response to reviewers

The authors thank the academic editor and the reviewers for the revision of this manuscript.

Points raised by the academic editor: 

- Regarding style requirements and file names: Citation brackets have been changed to square brackets and references to supplementary files have been updated (lines 247, 248, 269, 274). Supplementary captions as well as file names have been adapted accordingly and manuscript files have been renamed. Level 2 headings have been adapted (bold).

- Regarding ORCID iD for the corresponding author: The ORCID iD of the corresponding author has been added. Information regarding the corresponding author have been updated according to the author guidelines. 

Points raised by the academic editor after the first revision: owner informed consent

Lines 129-133: The paragraph was modified as follows: Horse owners were recruited for the above mentioned health surveillance study by the Franches-Montagnes and Swiss Warmblood breeding associations in collaboration with the ISME equine clinic [31,32]. They gave verbal consent if they were willing to take part in the study and agreed to a blood sample being taken from their horses. Sample collection has been approved by the ethic committee of the Swiss Canton of Vaud (VD2227.0, VD2227.1 and VD2227.2).”

The recruitement of cases and blood sample collections took place from 2005 until 2007. At that time it was common for such studies to ask only for verbal consent from the owners. The owners / handlers (appointed by the respective owner to present the horse) were present at the time of the clinical examination and the blood sample collection. If an owner / handler denied to take part in the study, the horse was automatically excluded.

Points raised by reviewer 1:

Reviewer 1 had concerns about the technical soundness of the manuscript, which refers to the inclusion criteria of the study subjects. 

Answer: Please find our comments below (major concern of reviewer 1).

Reviewer 1 stated that the manuscript is not presented in an intelligible fashion and written in standard English. 

Answer: Before our first submission, the manuscript was edited by a professional proofreading and editing service by a native speaker. We have sent the revised manuscript for a second proofreading and editing and hope that this addresses sufficiently the concerns of reviewer 1. The whole manuscript was edited and changes can be followed in the file named “Revised Manuscript with Tracked Changes”

Major concern of reviewer 1: Secondly, the authors evaluate the prognostic role of miRNA in predicting sarcoid recurrence/de novo occurrence including a reasonable number of horses with a long-term follow up. With respect to this section of the study, the inclusion criteria may have significantly affected the obtained results. While the diagnosis of sarcoid subtypes is clinical, the gold standard for the diagnosis of ES is histopathology. Cases included in the study were diagnosed by clinical examination while cases with an uncertain clinical diagnosis were excluded. As the sensitivity for the clinical diagnosis, as correctly reported by the authors, is about 80%, there is a potential up-to-20% of horses that do not have sarcoids: this may significantly alter the results of the study. Also, I understand that the improved protocol developed in 2017 was not applied to diagnose ES in this study as the last round of first examination was performed in 2013 according to Fig. 1. Is this correct? Since these miRNA biomarkers may then be used for future research it is crucial to include cases with a gold-standard-diagnosis. 

Answer: We have to acknowledge that that the lack of histopathological confirmation of ES diagnosis is a limitation of our study. Diagnosis of ES in our study was clinical, even though a definite diagnosis of ES requires histopathology. However, clinical diagnosis has a good reliability, particularly in the light of the fact, that we excluded all cases with an equivocal clinical diagnosis. Furthermore, we applied an improved diagnostic protocol for all follow up examinations from 2017 onwards. Before 2017, such a protocol was not yet available and could not be used for the initial clinical examinations and the follow-up examinations performed before 2017 (Haspeslagh M, Gerber V, Knottenbelt DC, Schüpbach G, Martens A, Koch C. The clinical diagnosis of equine sarcoids—Part 2: Assessment of case features typical of equine sarcoids and validation of a diagnostic protocol to guide equine clinicians in the diagnosis of equine sarcoids. Vet J. 2018 Oct 1;240:14–8). Because trauma of any nature carries the risk of exacerbating ES, biopsies were not performed in the cases included in this study. Furthermore, surgical excision of the relatively mild lesions in our study was not indicated in any of the horses at the time of the first presentation. Despite the lack of histopathological confirmation of ES cases, the strength of our study lies in the fact that the included study subjects showed a typical presentation and had a long-term follow-up. Since Koch et al. 2018 (Koch C, Martens A, Hainisch EK, Schüpbach G, Gerber V, Haspeslagh M. The clinical diagnosis of equine sarcoids — Part 1: Assessment of sensitivity and specificity using a multicentre case-based online examination. Vet J. 2018; 242: 77-82) had shown that atypical cases predominantly lead to misclassification and that Cases with typical features were significantly more likely to be assessed correctly, we are thus confident of the reliability of diagnosis in our cohort.

In a further (still unpublished and therefore not referenced) longitudinal experiment, we collected samples prospectively only from horses with a histopathological confirmation. We achieved similar results as in the present study: eca-miR-127 turned out as diagnostic biomarker for ES disease and none of the tested miRNAs was able to predict regression or progression of ES disease. Furthermore, we could confirm the influence of sex on miRNA expression.

In order to address this major concern in the manuscript, we included the following statement in the discussion (line 379-385): One limitation in this study is the lack of histopathological confirmation of ES disease, which is considered a diagnostic gold standard [44]. At the time of initial examination, biopsies were not performed since they represent a risk for disease aggravation, as any kind of trauma. Furthermore, in none of the horses, a surgical excision of the relatively mild ES lesions was indicated. In order to offset the lack of histopathological confirmation, only cases with a typical clinical presentation were included, and since its availability in 2017, an improved diagnostic protocol was additionally applied. Since it has been previously demonstrated that atypical cases predominantly lead to mis-classification, the reliability of diagnosis in our study cohort was deemed high [9]

The following citation [9] was added: Koch C, Martens A, Hainisch EK, Schüpbach G, Gerber V, Haspeslagh M. The clinical diagnosis of equine sarcoids — Part 1: Assessment of sensitivity and specificity using a multicentre case-based online examination. Vet J. 2018; 242: 77-82

Minor concerns of reviewer 1:

Line 94: sentence incomplete?

Answer: We deleted "For the CH-WB horses".

Line 368-9: rephrase

Answer: We corrected this sentence: " The samples utilized in this study were derived from a population-based biobank built up over years."

Line 245: provide reference

Answer: "A flow diagram depicting the recruitment of FM horses has been previously published [11] and further information is given in S1 Fig." The reference [11] has been added: Berruex F, Gerber V, Wohlfender FD, Burger D, Koch C. Clinical course of sarcoids in 61 Franches-Montagnes horses over a 5–7 year period. Vet Q. 2016 Oct;36(4):189–96. 

Points raised by reviewer 2:

Line 41 and others: please put a space bar between text and brackets. 

Answer: A space bar has been added between text an brackets in lines: 39, 41, 42, 44, 47, 52, 59, 61, 63, 67, 68, 70, 72, 74, 76, 79, 83, 92, 103, 107, 121, 129, 140, 147, 149, 151, 178, 188, 196, 235, 271, 374, 375, 388, 391, 393, 398, 401, 402, 406, 408, 410, 413, 420, 422, 425, 428, 432, 434, 437, 439, 441, 444, 453, 455, 459, 462, 463, 465, 472, 474, 478, 486, 490.

Line 94: maybe a typo

Answer: We deleted "For the CH-WB horses".

Line 108: were control horses of the same breed of the studied ones? And of the same age? Please specify the inclusion criteria for control cases.

Answer: The control horses were of the same age and same breeds as the ES cases. Additionally, we paid attention to reach a similar sex distribution in the equine sarcoid vs. control groups. For more detailed information please have a look at Supplementary Table 2. 

We modified the sentence in line 108 accordingly: Age matched controls were randomly selected from the large pool of tumor-free horses. Attention was paid to reach a similar breed and sex distribution in the ES vs. CTL group (S2 Table).

Why didn't the authors took in consideration the histotype of the sarcoids? Even if not so relevant from a clinical point of view, it would be interesting to evaluate is any difference could be present among different histotypes and relative miRNAs.

Answer: We agree that looking at the different types of sarcoids would have been interesting. However, in this study, the differentiation of ES types would have significantly reduced the power of our statistical analyses. For such an approach, a higher number of cases would have been needed. Furthermore, in a previous study the ES type did not have any influence on the expression of the tested miRNAs (Unger L, Abril C, Gerber V, Jagannathan V, Koch C, Hamza E. Diagnostic potential of three serum microRNAs as biomarkers for equine sarcoid disease in horses and donkeys. J Vet Intern Med 2021;35(1): 610-619).

Additional comment:

We noticed a typo in Table 2 (lines 354-357). The specificity of eca-miR-432 was corrected to 75% (95% CI = 43-95%).

---

## [Decision Letter · Decision Letter 1]

24 Nov 2021

Diagnostic and prognostic potential of eight whole blood microRNAs for equine sarcoid disease

PONE-D-21-19643R1

Dear Dr. Cosandey,

We’re pleased to inform you that your manuscript has been judged scientifically suitable for publication and will be formally accepted for publication once it meets all outstanding technical requirements.

Kind regards,

Silvia Sabattini

Academic Editor

PLOS ONE

Additional Editor Comments (optional):

Reviewers' comments:

Reviewer's Responses to Questions

**Comments to the Author**

1. If the authors have adequately addressed your comments raised in a previous round of review and you feel that this manuscript is now acceptable for publication, you may indicate that here to bypass the “Comments to the Author” section, enter your conflict of interest statement in the “Confidential to Editor” section, and submit your "Accept" recommendation.

Reviewer #2: All comments have been addressed

2. Is the manuscript technically sound, and do the data support the conclusions?

Reviewer #2: Yes

3. Has the statistical analysis been performed appropriately and rigorously? 

Reviewer #2: Yes

4. Have the authors made all data underlying the findings in their manuscript fully available?

Reviewer #2: Yes

5. Is the manuscript presented in an intelligible fashion and written in standard English?

Reviewer #2: Yes

6. Review Comments to the Author

Reviewer #2: The main study concern has been properly discussed both in the review letter and in the manuscript, and all the style suggestions addressed. Even with an important technical missing, authors take in consideration previous publications that can support their results. The manuscript technically sounds and It is now recommended for the publication.

7. PLOS authors have the option to publish the peer review history of their article (what does this mean?). If published, this will include your full peer review and any attached files.

Reviewer #2: No

---

## [Editor Report · Acceptance letter]

14 Dec 2021

PONE-D-21-19643R1 

Diagnostic and prognostic potential of eight whole blood microRNAs for equine sarcoid disease 

Dear Dr. Cosandey:

I'm pleased to inform you that your manuscript has been deemed suitable for publication in PLOS ONE. Congratulations! Your manuscript is now with our production department. 

Kind regards, 

on behalf of

Dr. Silvia Sabattini 

Academic Editor

PLOS ONE